# High-Specificity Test Algorithm for Bovine Tuberculosis Diagnosis in African Buffalo (*Syncerus caffer*) Herds

**DOI:** 10.3390/pathogens11121393

**Published:** 2022-11-22

**Authors:** Charlene Clarke, Netanya Bernitz, Wynand J. Goosen, Michele A. Miller

**Affiliations:** 1DSI-NRF Centre of Excellence for Biomedical Tuberculosis Research, SAMRC for Tuberculosis Research, Division of Molecular Biology and Human Genetics, Faculty of Medicine and Health Sciences, Stellenbosch University, Cape Town 7505, South Africa; 2Faculty of Infectious and Tropical Diseases, London School of Hygiene and Tropical Medicine, Keppel Street, London WC1 7HT, UK

**Keywords:** African buffaloes, bovine tuberculosis, IGRA, IP-10 release assay, *M. bovis*, nontuberculous mycobacteria, serial testing, specificity, *Syncerus caffer*

## Abstract

Ante-mortem bovine tuberculosis (bTB) tests for buffaloes include the single comparative intradermal tuberculin test (SCITT), interferon-gamma (IFN-γ) release assay (IGRA) and IFN-γ-inducible protein 10 release assay (IPRA). Although parallel test interpretation increases the detection of *Mycobacterium bovis* (*M. bovis*)-infected buffaloes, these algorithms may not be suitable for screening buffaloes in historically bTB-free herds. In this study, the specificities of three assays were determined using *M. bovis*-unexposed herds, historically negative, and a high-specificity diagnostic algorithm was developed. Serial test interpretation (positive on both) using the IGRA and IPRA showed significantly greater specificity (98.3%) than individual (90.4% and 80.9%, respectively) tests or parallel testing (73%). When the SCITT was added, the algorithm had 100% specificity. Since the cytokine assays had imperfect specificity, potential cross-reactivity with nontuberculous mycobacteria (NTM) was investigated. No association was found between NTM presence (in oronasal swab cultures) and positive cytokine assay results. As a proof-of-principle, serial testing was applied to buffaloes (n = 153) in a historically bTB-free herd. Buffaloes positive on a single test (n = 28) were regarded as test-negative. Four buffaloes were positive on IGRA and IPRA, and *M. bovis* infection was confirmed by culture. These results demonstrate the value of using IGRA and IPRA in series to screen buffalo herds with no previous history of *M. bovis* infection.

## 1. Introduction

African buffaloes (*Syncerus caffer*) are important maintenance hosts of bovine tuberculosis (bTB) in South Africa [1]. The primary causative agent of bTB in buffaloes is *Mycobacterium bovis* (*M. bovis*)*,* a pathogenic member of the *Mycobacterium tuberculosis* complex (MTBC) [2]. However, other members of the MTBC have also been shown to cause TB in these bovids, including *M. orygis* [3] and *M. tuberculosis* [4]. Control measures in South Africa require mandatory screening for *M. bovis* infection prior to the movement of buffaloes [1]. Since test-positive animals are usually culled, and the farm placed under quarantine until evidence can be provided that *M. bovis* has been eradicated from the herd [1], it is crucial to use accurate diagnostic tests to identify all infected animals without the unnecessary culling of false-positive buffaloes [5]. A herd is considered eradicated from disease if all animals in the herd have two consecutive negative tuberculin skin test (TST) results [6,7]. 

To understand the diagnostic accuracy of a test, it is important to assess sensitivity and specificity, i.e., the ability to correctly distinguish truly infected or uninfected animals [8]. Ideally, these parameters should be close to 100% for optimal diagnostic performance, but with an inverse relationship between sensitivity and specificity for an imperfect test, increasing test sensitivity results in compromised specificity, and vice versa [8]. Various factors influence bTB test sensitivity and specificity when using methods based on immune responses, including the test antigens selected for the assay, exposure of the host to environmental mycobacteria, and the specific host biomarkers measured in cytokine release assays (CRA) [2,8]. 

Ante-mortem diagnosis of *M. bovis* infection in buffaloes relies on the measurement of antigen-specific cell-mediated host-immune responses [2,9]. These include the single comparative intradermal tuberculin test (SCITT), QuantiFERON^®^-TB Gold Plus (QFT) or Bovigam^®^ interferon-gamma (IFN-γ) release assays (IGRA) and QFT IFN-γ inducible protein 10 release assay (IPRA) [10]. A number of QFT IGRAs have been described for bTB diagnosis in African buffaloes, including an in-house bovine IFN-γ enzyme-linked immunosorbent assay (ELISA) [11], the commercially available Mabtech bovine IFN-γ ELISA PRO [12] and Cattletype^®^ IFN-γ ELISA (recently discontinued by manufacturer) [13]. Serological assays, which measure humoral immune responses to *M. bovis* antigens, have been shown to have low sensitivity for the early immunological detection of *M. bovis* infection in buffaloes [10,14]. In bovids, these assays are more likely to be positive in animals with disease; therefore, bTB detection in buffaloes relies on primarily CMI assays for early detection of infection [10,14].

The TST and Bovigam^®^ IGRA measure host responses to purified protein derivatives (PPD), which are antigens consisting of a crude mixture of proteins, carbohydrates and lipids obtained from *M. bovis* (bovine PPD), *M. avium* (avian PPD), and in some cases from *M. fortuitum* (fortuitum PPD) [2,8,15]. Assays using PPDs typically have high sensitivity due to the greater number of antigens used to stimulate a detectable immune response, but often have compromised specificity due to cross-reactive responses to environmental mycobacteria that share antigens with MTBC [2,9,16]. The single intradermal test measures the host response to bovine PPD. However, the inclusion of avian PPD in the SCITT has led to increased specificity by accounting for sensitisation to *M. avium* and closely related nontuberculous mycobacterial (NTM) species [17]. In this test, the host immune response to bovine PPD is measured as a change in skin-fold thickness at 72 hours. This is compared to the change in skin-fold thickness in response to avian PPD. The difference in these results is then used to determine the test result of the buffalo [16]. Similarly, the inclusion of fortuitum PPD has also been shown to increase test specificity in PPD-based IGRAs [15]. Specificity can be further improved by using MTBC specific antigens, early secretory antigen 6 kDa (ESAT-6) and culture filtrate protein 10 (CFP-10), in the stimulation platform for CRAs [9,13]. This, however, comes at the expense of test sensitivity [13]. The inclusion of additional biomarkers, such as IP-10, which is released in significantly greater concentrations than IFN-γ, has been shown to improve sensitivity [13]. However, genes encoding for ESAT-6 and CFP-10 antigens are shared with some NTM species, leading to cross-reactive immunological responses and false-positive IGRA and IPRA results [17,18]. Therefore, test interpretation requires knowledge of potential confounders, such as NTM exposure of the host. 

Although current guidelines for testing buffaloes in South Africa rely on the SCITT, the use of parallel testing (i.e., test-positive if positive on any bTB test included in the test algorithm) has been shown to improve the detection of infected animals [8,13,19]. With increased sensitivity, there is a trade-off in specificity, resulting in the culling of some false-positive buffaloes, which is of lesser concern when attempting to eradicate bTB in a herd [13]. However, this approach is less suitable for bTB testing of historically bTB-free African buffalo herds, especially those with high economic or genetic value since each false-positive buffalo that is culled leads to significant loss. In these scenarios, a testing algorithm with high specificity is needed to minimise unnecessary culling. However, this fit-for-purpose high-specificity algorithm has not been evaluated in African buffaloes. Previous studies, to determine bTB test specificity values, used buffaloes that were *M. bovis* culture-negative, but from exposed herds (bTB endemic herds); which may result in an inaccurate estimate of specificity [19]. Therefore, this study aimed to (1) determine individual specificities of the QFT Mabtech IGRA, QFT IPRA, and SCITT in known *M. bovis* unexposed African buffaloes, (2) evaluate test algorithms to optimise specificity in these *M. bovis* unexposed buffalo herds, (3) determine if there is an association between false-positive bTB test results in *M. bovis* unexposed herds and presence of NTMs, and (4) apply the high-specificity testing algorithm to an independent historically bTB-free herd as a proof of principle. 

## 2. Materials and Methods

### 2.1. Animals and Sample Collection

Samples were collected between 2017 and 2021 from African buffaloes (n = 268) at historically bTB-free private game farms in South Africa (Limpopo, northwest and northern Cape provinces). These farms were regarded as historically bTB-free based on negative TST results during periodic testing. Any TST-positive buffaloes were culled, and mycobacterial culture and speciation performed on tissue samples. However, *M. bovis* has not been detected to date in these animals. Following the capture and immobilisation of buffaloes, whole blood was collected by jugular venipuncture into lithium heparin tubes, as previously described [11]. The SCITT was performed after blood collection. Within eight hours of sample collection, whole blood stimulation was performed using the QuantiFERON^®^-TB Gold Plus (QFT; Qiagen, Hilden, Germany) platform. Briefly, 1 mL heparinised whole blood was aliquoted into each of the QFT tubes, which included the antigen tube containing cluster of differentiation (CD) 4 and CD8 T-cells stimulating ESAT-6 and CFP-10 antigens (QFT TB2 tube), the negative control containing saline (QFT Nil), and a positive control (QFT mitogen) containing phytohaemagglutinin (PHA), which was additionally spiked with 10 µg/mL pokeweed mitogen (Sigma Aldrich, St. Louis, MO, USA) to ensure stimulation. The fourth tube (QFT TB1) only contains CD4 stimulating antigens. Following 100% agreement in test outcomes from TB1 and TB2 tubes in a previous study [19], the TB1 tube was not included for these buffaloes. After inverting all tubes 10 times, tubes were incubated at 37 °C for 24 h, prior to harvesting plasma fractions by centrifugation at 3000 rpm for 15 min.

Cohort 1 consisted of 115 buffaloes from three historically bTB-free farms tested in 2021. The QFT Mabtech IGRA and QFT IPRA were performed on all buffaloes, and SCITTs were performed on a subset (n = 69). Oronasal swabs were collected in saline from all buffaloes for mycobacterial culture. This cohort was used to calculate assay specificity. 

Cohort 2 consisted of 153 buffaloes from one historically bTB-free farm (different location to Cohort 1). A single SCITT reactor buffalo bull was detected during bTB testing for movement for sale, according to regulations by the South African Department of Agriculture, Land Reform and Rural Development (DALRRD). Following this, the entire herd was tested in 2017, using QFT Cattletype^®^ IGRA and QFT IPRA. The owner consented to have the buffaloes that had positive results on both assays culled, and tissue samples were collected during post-mortem examination for mycobacterial culture. 

The Cattletype^®^ IFN-γ ELISA was discontinued before Cohort 1 buffaloes were tested, and therefore the Mabtech IFN-γ ELISA was used. A comparison of test performance between these assays showed no significant differences [12]. Therefore, the assay specificities calculated using results from Cohort 1 buffaloes were used to retrospectively evaluate the high-specificity test algorithm using results from Cohort 2 buffaloes. 

### 2.2. Single Comparative Intradermal Tuberculin Tests

Single comparative intradermal tuberculin tests (SCITT) were performed following immobilisation on a subset of Cohort 1 buffaloes (n = 69), and 1 buffalo from Cohort 2, as previously described [11]. A buffalo-specific cut-off value of 3 mm was used to identify SCITT-positive buffaloes [16]. 

### 2.3. Cytokine Release Assays

Cytokine concentrations in plasma from QFT-stimulated whole blood samples were measured with the bovine IP-10 ELISA (Kingfisher Biotech, St Paul, MN, USA) in all buffaloes, as previously described [20] and with commercially available bovine IFN-γ ELISAs (IGRA); the bovine IFN-γ ELISA Pro assay (Mabtech, Stockholm, Sweden) was used for Cohort 1 buffaloes, whereas the Cattletype^®^ Ruminant IFN-γ ELISA (Indical Bioscience, Leipzig, Germany) was used with Cohort 2 animals, as previously described [12]. Mycobacterial antigen-specific cytokine concentrations were determined by subtracting QFT Nil cytokine concentrations from QFT TB2 cytokine concentrations. Buffalo-specific assay cut-off values have previously been described; buffaloes were regarded as IPRA-positive if the antigen-specific IPRA result was ≥ 1486 pg/ml [20], Mabtech IGRA-positive if the concentration was ≥8 pg/ml and Cattletype^®^ IGRA-positive if the relative signal to positive ratio (S/P%) was ≥5% [12]. 

### 2.4. Bovine TB Test Interpretation

The parallel interpretation of the IGRA and IPRA was considered test-positive if the result was positive for either IGRA or IPRA (negative if both IGRA and IPRA negative), whereas a positive serial test result was assigned if both IGRA and IPRA were positive, or positive if all three tests were positive (based on IGRA, IPRA and SCITT). 

### 2.5. Mycobacterial Culture and Speciation

Oronasal swab samples (Cohort 1) and tissue samples from culled buffaloes (Cohort 2) were processed for mycobacterial culture in the biosafety level 3 (BSL-3) laboratory at Stellenbosch University. A 500 µL aliquot of each oronasal swab saline media, or approximately 1 cm^3^ of homogenised tissue (head, peripheral, abdominal and thoracic lymph nodes, lung) was cultured using the BACTEC^TM^ MGIT^TM^ 960 TB System (BD, Franklin Lakes, NJ, USA), as previously described [21,22]. Samples were regarded as culture-negative if no growth was detected after 56 days of incubation; no downstream analysis was performed on these samples. Culture-positive samples were subjected to inactivation at 99 °C for 30 min before removal from the BSL-3 facility for speciation. 

Mycobacterial speciation was based on polymerase chain reaction (PCR) amplicon sequences. Oronasal swab cultures from buffaloes in Cohort 1 were screened for highly conserved RNA polymerase beta subunit (*rpoB*) and heat-shock protein 65 kDa (*hsp*65) gene targets, using previously described primers for amplification [23,24] in a Veriti^TM^ 96-Well Thermal Cycler (Applied Biosystems, Waltham, MA, USA). The preparation of PCRs, cycling conditions, and gel electrophoresis were performed as previously described [22]. All PCR amplicons that showed bands on gel electrophoresis were sent to the Central Analytical Facility (CAF), Stellenbosch University, for Sanger sequencing (Illumina, Inc., San Diego, CA, USA). BioEdit Biological Sequence Alignment Editor (BioEdit (RRID:SCR_007361) was used to generate sequence contigs, which were blasted on NCBI Basic Local Alignment Search Tool for nucleotides (BLASTn) to find sequence matches in the NCBI database. The presence of mycobacterial species was confirmed if sequences from either *hsp*65 or *rpoB* matched with mycobacterial species in the NCBI database, with an identity match ≥90%. The NTM species with the highest percentage match was selected as the result for each buffalo. *Mycobacterium avium* complex (MAC) classification was followed as described by To et al. [25]. Mycobacterial cultures from tissue samples collected from culled Cohort 2 buffaloes during post-mortem examination were speciated by region of difference (RD) PCR, as described by Warren et al. [26]. 

### 2.6. Data Analyses

Buffaloes were considered positive if positive on either IGRA or IPRA, using the parallel test interpretation, whereas serial test interpretation regarded buffaloes as positive if positive on IGRA, IPRA and SCITT (if included). The specificities for individual tests and test combinations of the Mabtech IGRA, IPRA, and SCITT (in series and in parallel) were calculated based on the percentage of *M. bovis* unexposed buffaloes from historically bTB-free herds (Cohort 1) that were test-negative on each test, using an online tool (MedCalc Software Ltd 2022; Ostend, Belgium). Since buffaloes originated from historically bTB-free herds, animals were regarded as true-negative if they were test-negative (on IGRA, IPRA or SCITT), and false-positive if test-positive (on IGRA, IPRA or SCITT). Proportions of test-positive and -negative buffaloes were compared using the exact binomial McNemar’s test with Bonferroni correction for multiple comparisons (GraphPad Software, Inc., 2022; online calculator; https://www.graphpad.com/quickcalcs/mcNemar2/; San Diego, CA, USA). Contingency tables were used to determine the association between NTM presence or absence in oronasal swab cultures and the Mabtech IGRA, IPRA and SCITT test results (Cohort 1), using the Fisher’s exact test. Differences in cytokine responses between animals with NTMs present or absent were compared using the nonparametric Mann-Whitney U test, following the D’Agostino–Pearson normality test. Results were considered statistically significant if *p* < 0.05. All statistical tests were performed on GraphPad Prism version 7 software (GraphPad Software Inc., San Diego, CA, USA). 

## 3. Results

Specificity values were calculated for three assays (Mabtech IGRA, IPRA and SCITT), individually and applied in parallel and in series (Table 1), using blood test results from African buffaloes on historically bTB-free farms (Cohort 1). The SCITT (individually and in series with IGRA and IPRA) identified all 69 buffaloes as uninfected, with a calculated specificity of 100%. The specificities of the individual CRAs (Mabtech IGRA and IPRA) were 90.4% and 80.9%, respectively, although the difference was not statistically significant (*p* > 0.06). Parallel interpretation of the Mabtech IGRA and IPRA resulted in a decrease in specificity, to 73.0%. When a serial interpretation was applied to Mabtech IGRA and IPRA results, there was high specificity, at 98.3%, which was significantly greater than the specificities of the individual (*p* < 0.008) or parallel interpretations (*p* < 0.0001) of the cytokine release assays. 

Mycobacterial culture results of oronasal swabs from buffaloes in Cohort 1 (n = 115) were used to assess an association between Mabtech IGRA and IPRA results and the presence of mycobacteria. No MTBC organisms were detected; however, NTMs were identified in 69 buffalo oronasal swabs, with 34 identified to species level based on *hsp*65 and *rpoB* amplicon Sanger sequencing and the other 35 samples identified as belonging to the *Mycobacterium* genus. There were no significant associations between NTM presence or absence and Mabtech IGRA-positive or -negative results (*p* = 0.5), or IPRA-positive or -negative results (*p* = 0.15), using the Fisher’s exact test (Table 2). There were also no significant differences in antigen-specific IFN-γ (*p* = 0.51) or IP-10 (*p* = 0.32) concentrations in buffaloes with NTMs present compared to those animals which did not have NTMs detected (Figure 1). 

Since NTM species could be identified in 34 buffalo oronasal swabs, these were grouped according to whether the species had been shown to contain the RD1 region [17,27]. The presence of RD1-containing NTM species was not associated with a positive Mabtech IGRA or IPRA result in these buffaloes (*p* > 0.05; data not shown). The largest number of NTM species matches were *Mycobacterium avium* complex (MAC; 17 matches). However, there was also no association between the presence of MAC and a positive IPRA or IGRA result (*p* > 0.05; data not shown).

Based on the high specificity of the combined Mabtech IGRA and IPRA when used in series, this algorithm was selected for retrospective evaluation of results from an unrelated historically bTB-free buffalo herd (n = 153; Cohort 2). Although the Cattletype^®^ IGRA was used for Cohort 2 buffaloes, Cattletype^®^ and Mabtech IGRAs had similar test performances [12]. A total of 28 of 153 buffaloes (18%) had positive IPRA results but were negative on Cattletype^®^ IGRA. Using the serial interpretation, these animals were consdered test-negative. Four buffaloes (3%) were positive for both Cattletype^®^ IGRA and IPRA and considered test-positive. These four individuals had been culled due to suspicion of being infected, and post-mortem examinations were performed. All four buffaloes were confirmed to be infected with *M. bovis* based on mycobacterial culture of tissues and speciation by RD PCR. The remaining 124 buffaloes had negative results for both Cattletype^®^ IGRA and IPRA. 

## 4. Discussion

In this study, serial testing with Mabtech IGRA and IPRA (with or without SCITT included) showed high specificity compared to individual test specificity or parallel interpretation of test results when used in *M. bovis* unexposed African buffaloes. As a proof of principle, this high-specificity testing algorithm, i.e., serial interpretation of IGRA and IPRA results, was applied to an independent historically bTB-free buffalo herd. The four buffaloes that were test-positive (positive on both Cattletype^®^ IGRA and IPRA) were confirmed *M. bovis*-infected based on mycobacterial culture and speciation, demonstrating that CRA serial test interpretation may have promise for screening buffaloes with no previous history of bTB, since no false test-positive buffaloes were culled. Following the detection of CRA reactor animals in Cohort 1, the presence of mycobacteria and its association with positive CRA results were investigated. Although only NTMs were detected in oronasal secretions, their presence showed no clear association with a positive CRA result.

The diagnostic accuracy of an assay is primarily defined by its specificity and sensitivity [8]. These parameters are often considered constant across different populations [8]. However, sensitivity and specificity may be influenced by various factors, including infection stage and disease severity, and exposure to environmental mycobacteria [2,8]. Furthermore, the infection status of the herd is an important consideration for the optimal application of a diagnostic assay or algorithms. Buffalo herds that are endemic for bTB usually require tests with high sensitivity, since the aim is to identify as many *M. bovis*-infected buffaloes as possible, at the expense of culling some false-positive animals [13]. A herd that is historically bTB-free or has a low bTB prevalence requires a test with high specificity, which will correctly identify *M. bovis*-uninfected buffaloes while minimising the removal of false-positive animals [8,28]. 

In this study, we aimed to determine which individual or combination of tests would have the highest specificity for screening buffaloes from historically bTB-free farms. The true specificity of bTB tests in buffaloes is often not accurately calculated due to a lack of samples from truly *M. bovis*-uninfected, -unexposed buffalo herds, especially based on the gold standard of mycobacterial culture of tissue [12]. Therefore, specificity is sometimes determined using *M. bovis*-uninfected, but -exposed animals [13]. However, mycobacterial culture is an imperfect gold-standard test, since some *M. bovis*-infected buffaloes may have negative culture results, thereby resulting in a decreased assay specificity [12,13]. 

The specificity values of the Mabtech IGRA, IPRA, and SCITT were calculated in a cohort of buffaloes that had no history of exposure to *M. bovis,* based on previous herd test results. Buffaloes from these herds had undergone sporadic culling and no *M. bovis* had been detected by mycobacterial culture of tissue samples [22]. Although ante-mortem culture has limited sensitivity, no MTBC members were detected in the oronasal secretions of the unexposed buffaloes used to calculate specificity. In these buffaloes, the Mabtech IGRA had a higher (but not significantly) specificity (90.4%) than IPRA (80.9%). This result was expected since IP-10 is known to be a more sensitive cytokine biomarker compared to IFN-γ [13]. The specificity for IPRA in the current study was higher than that reported by Bernitz et al. [19] at 67%. However, Bernitz et al. [19] determined specificity in *M. bovis* culture-negative buffaloes from a bTB-endemic herd. In a small group of *M. bovis*-unexposed buffaloes, IPRA specificity was calculated as 100% [13], and Mabtech IGRA and Cattletype^®^ IGRA as 94% and 97%, respectively [12]. Surprisingly, in the current study, SCITT had the highest specificity at 100%. Previous studies have shown that the SCITT has decreased specificity when compared to assays using ESAT-6 and CFP-10 antigens [2,16]. Parallel interpretation of the Mabtech IGRA and IPRA had a lower specificity (73%) compared to specificities of the individual tests, although it was higher than that previously reported for buffaloes (63% using the Cattletype^®^ IGRA and IPRA), which calculated specificity based on results from culture-negative buffaloes in a bTB endemic herd [19]. As expected, this value was lower compared to specificity previously calculated from unexposed herds (Cattletype^®^ IGRA and IPRA) at 100% [13]. Serial testing with Mabtech IGRA and IPRA showed significantly improved specificity at 98.3% (or 100% when SCITT was included) over individual CRAs and parallel testing and was therefore selected as the optimal high-specificity algorithm. Although SCITT had the highest specificity at 100%, whereas serial testing with only CRAs was 98.3%, it was decided not to include this test in further algorithm evaluations for various reasons. The CRAs provide a cost-and-welfare advantage over SCITT since a single immobilisation is required for blood collection, whereas SCITT requires that buffaloes are immobilised twice and are usually held captive for the three days in between immobilisation events [10,13]. In addition, CRAs allow the retesting of animals without a waiting period between bTB testing, whereas SCITT requires a waiting period between tests to avoid in vivo sensitisation [2,13]. Finally, SCITTs may be subjective to operator bias [8,13]. 

Some Mabtech IGRA or IPRA reactors were detected in the buffaloes used for the specificity calculations (Cohort 1), which resulted in decreased assay specificity. This decreased test specificity has been hypothesised to be due to cross-reactivity with environmental mycobacteria [17]. In the current study, oronasal swab culture results were used as a proxy for NTM exposure, based on mycobacterial culture and *hsp*65 or *rpoB* amplicon Sanger sequencing [22]. Members of the MTBC were not detected in any buffalo oronasal swab cultures. However, NTMs were detected in 60% of the buffaloes. Various studies have found that NTMs may impede diagnostic performance through cross-reactive immune responses to antigens shared between MTBC members and NTMs [9,17]. However, no association was found between the presence of NTMs and positive Mabtech IGRA or IPRA results. Nontuberculous mycobacteria that were previously described as containing the RD1 region, a genetic region containing virulent factors *esat*-6 and *cfp*-10 and associated with virulent MTBC members as well as some NTM species [9,29], showed no association with CRA results. The presence of *Mycobacterium avium* complex (MAC) members also did not appear to influence CRA results. Positive CRA results could also have been an indication of *M. bovis* infection, although MTBC was not detected in oronasal swab cultures. However, since oronasal swab cultures indicate the presence of NTMs in the oronasal cavities of the animals, it may not represent NTM infection. Other studies have also reported possible cross-reactivity with organisms other than NTMs, such as *Nocardia, Rhodococcus* and *Corynebacterium,* which are closely related to the *Mycobacterium* genus [2,30,31]. Therefore, the cause of false-positive CRA results should be investigated in future studies. 

As a proof-of-principle, the high-specificity test algorithm with CRAs showed good diagnostic performance when applied to a historically bTB-free buffalo herd (Cohort 2). Although the specificity calculation for QFT IGRA was performed using the Mabtech bovine IFN-γ ELISA Pro assay, the buffaloes in the test group used the Cattletype^®^ Ruminant IGRA. However, no significant differences in test performance between these assays have been reported [12]. The Cattletype^®^ IGRA was recently discontinued by the manufacturer, and therefore Mabtech IGRA was performed for Cohort 1 buffaloes (Cohort 2 buffaloes were tested before Cohort 1 animals). Even though a strict interpretation was used to define bTB test-positive individuals, four positive buffaloes were identified, which were confirmed to have *M. bovis* infection. Although some buffaloes were positive on IPRA, but not Cattletype^®^ IGRA, it is possible that these animals had early infection, since IP-10 is released at greater concentrations than IFN-γ [13,20]. However, infection could not be confirmed since these animals were not culled, due to the absence of a compensation program, and the herd was lost for follow-up. In this scenario, it is recommended that these buffaloes be quarantined and retested since it is expected that truly infected animals will eventually become IGRA-positive. This example demonstrates the importance of having a herd history for *M. bovis* infection to determine whether a high specificity or high sensitivity testing approach should be used [8,13]. 

Limitations of this study included the absence of gold-standard confirmation of the infection status of the buffaloes tested, using mycobacterial culture from tissue samples, especially those with a positive CRA result. Additionally, the use of oronasal swab cultures could not distinguish between NTM infection or contamination, which could impact whether the buffaloes developed immune responses to NTMs [32]. Due to availability, the bovine IFN-γ ELISAs differed between the buffalo herds used to calculate specificity (Mabtech IGRA) and the test group (Cattletype^®^ IGRA). 

In summary, this study calculated the specificities of Mabtech IGRA, IPRA, and SCITT as individual tests, interpreted in parallel and in series in known *M. bovis* unexposed buffaloes, which facilitated the development of a high-specificity test algorithm for use in bTB-free herds. Even in the presence of NTMs, the use of the IGRA and IPRA (with or without SCITT) in series yielded a high-specificity value, and accurately identified true *M. bovis-*infected buffaloes without culling high-value uninfected animals. These findings demonstrate that a serial test algorithm may have promise for screening buffalo herds with no previous evidence of bTB and should be assessed on a larger scale. 

## Figures and Tables

**Figure 1 pathogens-11-01393-f001:**
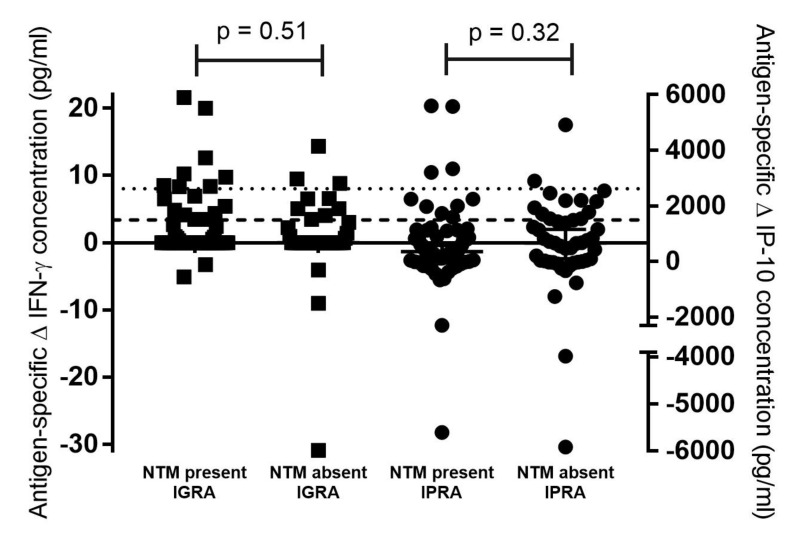
Concentrations of QuantiFERON^®^-TB Gold Plus (QFT) antigen-specific Mabtech interferon-gamma (IFN-γ, indicated by black squares) and interferon gamma inducible protein 10 (IP-10, indicated by black circles) in African buffalo samples from historically bovine tuberculosis-free farms (Cohort 1). Results are categorised based on whether nontuberculous mycobacteria (NTM) were present or absent in oronasal swab cultures, as determined by *hsp*65 or *rpoB* PCR amplicon Sanger sequencing. P-values for comparison of cytokine concentrations based on NTM presence or absence are indicated. Buffalo-specific cut-off values for Mabtech IGRA and IPRA are indicated as a dotted line or dashed line, respectively.

**Table 1 pathogens-11-01393-t001:** Calculated specificities of QuantiFERON^®^-TB Gold Plus interferon-gamma release assay (IGRA), interferon-gamma inducible protein 10 release assay (IPRA) and single comparative intradermal tuberculin test (SCITT) using African buffaloes from historically bovine tuberculosis-free farms (Cohort 1), based on individual, parallel and serial test interpretations. The numbers (and percentages) of false-positive and true-negative results for each test were based on previously determined buffalo-specific assay cut-off values.

Test	False Positive	True Negative	Specificity (95% Confidence Interval)
IGRA ^a^ (n = 115)	11 (9.6%)	104 (90.4%)	90.4% (83.5–95.1%)
IPRA ^b^ (n = 115)	22 (19.1%)	93 (80.9%)	80.9% (72.5–87.6%)
SCITT (n = 69)	0 (0%)	69 (100%)	100% (94.8–100%)
Parallel testing (IGRA ^a^, IPRA ^b^)	31 (27%)	84 (73%)	73.0% (64.0–80.9%)
Serial testing (IGRA ^a^, IPRA ^b^)	2 (1.7%)	113 (98.3%)	98.3% (93.9–99.8%)
Serial testing (IGRA ^a^, IPRA ^b^, SCITT)	0 (0%)	69 (100%)	100% (94.8–100%)

^a^ QuantiFERON^®^-TB Gold Plus stimulation platform with Mabtech bovine IFN-γ ELISA. ^b^ QuantiFERON^®^-TB Gold Plus stimulation platform with Kingfisher bovine IP-10 ELISA

**Table 2 pathogens-11-01393-t002:** The proportions (and percentages) of African buffaloes from historically bovine tuberculosis-free herds (Cohort 1) that had positive or negative test results using the QuantiFERON^®^-TB Gold Plus stimulation platform with cytokine release assays (interferon-gamma release assay (IGRA), interferon gamma inducible protein 10 release assay (IPRA)), categorised based on those that had nontuberculous mycobacteria (NTM) present or absent in oronasal swab cultures.

	NTM Present	NTM Absent	*p*-Value (Fisher’s Exact Test)
IGRA ^a^ test-positive	8/11 (73%)	3/11 (27%)	0.5
IGRA ^a^ test-negative	61/104 (59%)	43/104 (41%)	
IPRA ^b^ test-positive	10/22 (45%)	12/22 (55%)	0.15
IPRA ^b^ test-negative	59/93 (63%)	34/93 (37%)	

^a^ QuantiFERON^®^-TB Gold Plus stimulation platform with Mabtech bovine IFN-γ ELISA. ^b^ QuantiFERON^®^-TB Gold Plus stimulation platform with Kingfisher bovine IP-10 ELISA

## Data Availability

Correspondence and requests for materials should be addressed to Dr. Wynand Goosen.

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
