# Peer review of "High-Specificity Test Algorithm for Bovine Tuberculosis Diagnosis in African Buffalo (Syncerus caffer) Herds"

_pathogens, 2022, doi:10.3390/pathogens11121393_

Round 1

Reviewer 1 Report

A clear, well written paper describing a study undertaken to estimate the specificity of different ante-mortem tests (and test combinations) of cell-mediated host immune responses to Mycobacterium bovis in a small cohort of 115 African buffalo kept on three presumed TB-free herds private game farms in South Africa. The paper should be of interest to wildlife and zoo veterinarians, and researchers, diagnosticians, risk managers and regulators involved in the control/eradication of bovine/animal TB worldwide.  I could not identify any serious methodological flaws.  Therefore, I recommend acceptance of this paper for publication, subject to clarification of three relatively minor points, as follows:

1. M&M, Animals and sample collection (lines 110 to 113):  Can the authors elaborate a little more as to the evidence used to determine / conclude that the three buffalo/game farms in Cohort 1 were "historically bTB-free"?  E.g. was this based on their location, their lack of contact with other (potentially infected) herds of cattle or African buffalo, no live animals introduced from other herds (i.e. a closed-herd policy), any previous post-mortem and/or ante-mortem surveillance for TB conducted in those herds with consistently negative results, etc. and, if so, over how many years? 

Additionally, please clarify whether the 115 animals selected for cohort 1 and cohort 2 were all the buffalo present on the three game farms, or whether they only constituted a certain proportion of all the buffalo kept on those premises.  Is it possible to provide some descriptive statistics on the age and sex distribution of the sampled animals in each cohort?

2. Results (lines 252-257): This may be out of the scope of this study, but for completeness, it would have been useful to know the consequences and any follow-up actions arising from the confirmation of M. bovis infection in four IGRA (and IPRA)-positive animals from the historically presumed bTB-free herd of African buffalo in Cohort 2.  For instance, did the competent authorities undertake any forward tracings of any buffaloes that had been moved out of this game farm to other premises in SA and were those animals (or their herds of destination) tested for TB?  In light of those results, were any of the 25 animals positive on IPRA only (Note - the Abstract states n=28), culled as strong TB suspects, i.e. parallel re-interpretation of the test results following confirmation of M. bovis infection in the herd?

3. Discussion - somewhere in this section, could you comment on the potential role of antibody tests for TB screening of presumed free (and infected) herds of African buffalo?  Would the inclusion of a test for serum antibodies, in combination with the SICTT or IGRA/IPRA tests of cell-mediated immunity help increase the specificity of the testing algorithm in bTB-free herds, or the sensitivity in suspected/confirmed infected herds? Has this approach been attempted in this species?  Is this a pragmatic proposition?  If not, why not?

Author Response

11 November 2022

Response to reviewers and resubmission of manuscript ID pathogens-2014335: “High specificity test algorithm for bovine tuberculosis diagnosis in African buffalo (Syncerus caffer) herds”.

We would like to thank the reviewers for the opportunity to resubmit a revised version of our manuscript. We appreciate the constructive comments and have truly taken it into consideration and made the changes suggested. We have carefully revised the manuscript and hope we have now addressed the comments.

All responses to the reviewers are made below in a detailed point-by-point response, describing exactly what amendments have been made to the manuscript in track changes and where these can be found.

Yours, sincerely,

Wynand J. Goosen

Division of Molecular Biology and Human Genetics

Faculty of Medicine and Health Sciences

Stellenbosch University

Reviewer 1

Comment 1:

M&M, Animals and sample collection (lines 110 to 113):  Can the authors elaborate a little more as to the evidence used to determine / conclude that the three buffalo/game farms in Cohort 1 were "historically bTB-free"?  E.g., was this based on their location, their lack of contact with other (potentially infected) herds of cattle or African buffalo, no live animals introduced from other herds (i.e., a closed-herd policy), any previous post-mortem and/or ante-mortem surveillance for TB conducted in those herds with consistently negative results, etc. and, if so, over how many years?

Response:

Movement of African buffaloes between farms requires mandatory bovine TB (bTB) testing in South Africa, according to Department of Agriculture, Land Reform and Rural Development (DALRRD) policies. Buffaloes that test positive on the tuberculin skin test (TST) are immediately reported to DALRRD and typically culled. These private game reserves undergo regular bTB testing prior to buying or selling buffaloes. Any buffaloes that were TST positive in the past were culled, post-mortem examination performed, followed by mycobacterial culture and speciation on collected tissue samples. Mycobacterium bovis were not isolated and identified in any buffaloes at any time in the herd’s history; NTM species were, however, identified from these tissue samples, which were believed to have caused false-positive responses on the TST. These buffalo herds were therefore considered historically bTB-free according to DALRRD.

We added this to the materials and methods section, lines 114-117: “These farms were regarded as historically bTB-free based on negative TST results during periodic testing. Any TST-positive buffaloes were culled, and mycobacterial culture and speciation performed on tissue samples. However, M. bovis has not been detected to date in these animals.”

This is also described in the Discussion section, lines 329-332: “The specificity values of the Mabtech IGRA, IPRA, and SCITT, were calculated in a cohort of buffaloes that had no history of exposure to M. bovis, based on previous herd test results. Buffaloes from these herds had undergone sporadic culling and no M. bovis had been detected by mycobacterial culture of tissue samples”.   

Comment 2:

Additionally, please clarify whether the 115 animals selected for cohort 1 and cohort 2 were all the buffalo present on the three game farms, or whether they only constituted a certain proportion of all the buffalo kept on those premises.  Is it possible to provide some descriptive statistics on the age and sex distribution of the sampled animals in each cohort?

Response:

The buffaloes that were used in this study constituted a variable proportion of the total buffalo population on the farms (exact numbers were unknown).

Although this would have been an interesting addition, we did not include age and sex distribution of the buffaloes in the paper, since it does not contribute to the main message of the paper, i.e., to evaluate a high specificity bTB test algorithm for detection of M. bovis infected buffaloes from historically bTB-free herds. We therefore feel that adding age and sex distribution to the paper will take the focus away from the main aims, especially since the cohorts were selected by the owner for testing. However, we will consider investigating these descriptive statistics in our future papers.

Comment 3:

Results (lines 252-257): This may be out of the scope of this study, but for completeness, it would have been useful to know the consequences and any follow-up actions arising from the confirmation of M. bovis infection in four IGRA (and IPRA)-positive animals from the historically presumed bTB-free herd of African buffalo in Cohort 2.  For instance, did the competent authorities undertake any forward tracings of any buffaloes that had been moved out of this game farm to other premises in SA and were those animals (or their herds of destination) tested for TB?  In light of those results, were any of the 25 animals positive on IPRA only (Note - the Abstract states n=28), culled as strong TB suspects, i.e. parallel re-interpretation of the test results following confirmation of M. bovis infection in the herd?

Response:

We agree with the reviewer. Unfortunately, the buffaloes from Cohort 2 were lost to follow-up, and due to a lack of resources, the animal health authorities did not do any tracing or mandate culling of IPRA positive animals, since there is no compensation plan, and it was based on a research test. Therefore, we could not provide further information regarding this herd, as described in lines 395-397: “However, infection could not be confirmed since these animals were not culled, due to absence of a compensation program, and the herd has been lost to follow-up”.

We have changed the number of animals in line 277 from “n = 25” to “n = 28”.

Comment 4:

Discussion - somewhere in this section, could you comment on the potential role of antibody tests for TB screening of presumed free (and infected) herds of African buffalo?  Would the inclusion of a test for serum antibodies, in combination with the SICTT or IGRA/IPRA tests of cell-mediated immunity help increase the specificity of the testing algorithm in bTB-free herds, or the sensitivity in suspected/confirmed infected herds? Has this approach been attempted in this species?  Is this a pragmatic proposition?  If not, why not?

Response:

Serological assays have been shown to have low sensitivity for the early host immunological detection of M. bovis infection in buffaloes and are more useful for describing host humoral immunological responses due to established disease (Lyashchenko et al. 2020; Bernitz et al. 2021). Therefore, bTB detection in these bovids typically relies on assays of cell-mediated immunity (CMI) (Bernitz et al. 2021). Since the stage of infection in potentially naturally infected buffaloes is unknown, it is preferable to use CMI assays in parallel to increase sensitivity (Bernitz et al. 2018, 2019), while these tests used in series improves specificity, along with the specific antigen platform (Goosen et al. 2015; Bernitz et al. 2019).

For these reasons, we did not use serological assays for bTB detection in buffaloes.

We have added a section on serological assays in lines 63-68: “Serological assays, which measure humoral immune responses to M. bovis antigens, have been shown to have low sensitivity for early immunological detection of M. bovis infection in buffaloes. In bovids, these assays are more likely to be positive in animals with disease, therefore, bTB detection in buffaloes relies on primarily on CMI assays for early detection of infection.”

Reviewer 2 Report

Abstract

“. Although parallel test interpretation increases detection of  Mycobacterium bovis-infected buffaloes, these algorithms may not be suitable for screening buffaloes  in historically bTB-free herds.” Add a brief words why these algorithms may not be suitable for screening buffaloes in historically bTB-free herds?

Before using “M.” as an abbreviation of Mycobacterium, you should mention the word in full followed by its abbreviation within brackets.

“In this study, the specificities of three assays were determined using M. bovis unexposed herds, and a high specificity diagnostic algorithm developed.” Mention what was your reference gold standard or used method for measuring specificity of your assays?

“Serial test interpretation using the IGRA and IPRA showed significantly greater specificity (98.3%) than individual tests or parallel testing (73%). When the SCITT was added, the algorithm had a 100% specificity” illustrate what does you mean by serial test interpretation? Add the specificity of each of IGRA, IPRA and SCITT separately, so that the advantage of higher specificity of using these tests in that way is more clear than using each of these tests alone.

Mention the method used for determination of specificity in the abstract.

“No association was found between NTM presence (in oronasal swab cultures) and positive cytokine assay results.” What about NTM which may exist in other parts of the body?

Add more details about the used methodology within the abstract.

Mention what does you mean by “serial testing”?

What does you mean by “high-value buffaloes”.

“Buffaloes positive on a 22 single test (n=28) were regarded as test negative” what does you mean by single test? Do you mean either IGRA,  IPRA or SCITT? Or do you mean that you had repeated testing by the same test several times?

“and M. bovis infection was confirmed following culling” what was the method of confirmation?

“These results demonstrate the value of using IGRA and IPRA in series” illustrate what does you mean by “in series”?

Introduction

“The causative agent of bTB in buffaloes is Mycobacterium 31 bovis, a pathogenic member of the Mycobacterium tuberculosis complex (MTBC) [2]” mention that Mycobacterium bovis is the major cause of bovine tuberculosis, but  Mycobacterium tuberculosis was reported recently to cause tuberculosis in buffalo and cite this paper “Borham, M., Oreiby, A., ElGedawy, A., Hegazy, Y., Hemedan, A., & AlGaabary, M. (2022). Abattoir survey of bovine tuberculosis in tanta, centre of the Nile delta, with in silico analysis of gene mutations and protein–protein interactions of the involved mycobacteria. Transboundary and Emerging Diseases69(2), 434-450.”

“Since test-positive animals are usually culled, and the farm placed under quarantine until evidence can be provided that M. bovis has been eradicated from the herd [1], it is crucial to use accurate diagnostic tests to identify all infected animals without unnecessary culling of false-positive buffaloes [3].” Mention what is the evidence used in South Africa to consider that tuberculosis was eradicated from a farm after culling of infected cases and quarantine?

“The inclusion of avian PPD to these tests has led to increased specificity by accounting for sensitisation to M. 62 avium and closely related nontuberculous mycobacterial (NTM) species [12]” Do you mean the inclusion of avian PPD in a comparative way with other PPD or antigens? Illustrate. Do not use the full term of NTM after being abbreviated before.

“Although current guidelines for testing buffaloes in South Africa rely on the SCITT, the use of parallel testing has been shown to improve detection of infected animals [4,9,14]”  add more illustration about the parallel testing meant here?

“4) apply the high specificity testing algorithm to an unrelated historically bTB-free herd as a proof- of-principle.” What does you mean by the word unrelated?

Material and Methods

“Samples were opportunistically collected” illustrate what does you mean by opportunistically?

Illustrate how the historically bTB-free private game farms were confirmed to be free? Does they usually subjected to regular periodical testing?

Add more description about the geographical distribution of sampling areas.

“Cohort 1 consisted of 115 buffaloes from three historically bTB-free farms tested in 2021. The QFT Mabtech IGRA and QFT IPRA were performed on all buffaloes, and SCITTs were performed on a subset (n = 69).” Illustrate the chronological order of these tests,  did you performed SCITTs before, after or simultaneously with IGRA and IPRA?

Did you considered oronasal culture negative swaps to be true negative? As you now there are many other forms of TB rather than the respiratory form, and even NTM may affect other parts of the body, and this is may affect your specificity calculations.

Does the aim of dividing the animals to cohort 1 and 2, is to compare the performance of the tests in TB-free and TB-containing populations?  If so, add more details about TB-positive cases of cohort 2.

 Under “Data analysis” illustrate what does you mean by in series and in parallel?

Under “Data analysis, how could you determined NTM presence or absence, although culture negative results do not eliminate the presence of NTM in other regions of the body?

Results

Illustrate what is meant by parallel and serial testing, preferably within the material and methods section.

How could you identify false positive and true negative?

Discussion

Do you think that 4 positive cases of Cohort 2 are enough to finally judge your principle of the high specificity of IGRA and IPRA serial testing? I suggest mentioning that is a limitation for you study and further investigations are still required to confirm this principle.

Add an interpretation why the presence of NTM showed no clear association with a positive CRA result?

Author Response

10 November 2022

Response to reviewers and resubmission of manuscript ID pathogens-2014335: “High specificity test algorithm for bovine tuberculosis diagnosis in African buffalo (Syncerus caffer) herds”.

We would like to thank the reviewers for the opportunity to resubmit a revised version of our manuscript. We appreciate the constructive comments and have truly taken it into consideration and made the changes suggested. We have carefully revised the manuscript and hope we have now addressed the comments.

All responses to the reviewers are made below in a detailed point-by-point response, describing exactly what amendments have been made to the manuscript in track changes and where these can be found.

Yours, sincerely,

Wynand J. Goosen

Division of Molecular Biology and Human Genetics

Faculty of Medicine and Health Sciences

Stellenbosch University

Reviewer 2

Comment 1:

Abstract. “Although parallel test interpretation increases detection of Mycobacterium bovis-infected buffaloes, these algorithms may not be suitable for screening buffaloes in historically bTB-free herds.” Add a brief words why these algorithms may not be suitable for screening buffaloes in historically bTB-free herds?

Response:

The parallel interpretation of bTB tests maximises the detection of M. bovis infected buffaloes. However, since this test interpretation has a high assay sensitivity and decreased specificity, false-positive buffaloes are also detected. Herds that are endemic for bTB, and therefore have high infection prevalence, require a high sensitivity assay to eradicate disease in the herd, at the expense of false-positive results. However, applying high sensitivity assays to herds that are historically bTB-free, will result in unnecessary loss of buffaloes, through culling of false-positive animals.

We have discussed this in lines 92-100 in the introduction section: “Although current guidelines for testing buffaloes in South Africa rely on the SCITT, the use of parallel testing (i.e., test-positive if positive on any bTB test included in the test algorithm) has been shown to improve detection of infected animals. With increased sensitivity, there is a trade-off in specificity, resulting in culling of some false-positive buffaloes, which is of lesser concern when attempting to eradicate bTB in a herd. However, this approach is less suitable for bTB testing of historically bTB-free African buffalo herds, especially those with high economic or genetic value since each false-positive buffalo that is culled, leads to significant loss. In these scenarios, a testing algorithm with high specificity is needed to minimise unnecessary culling” and lines 315-317 in the discussion section of the paper: “Buffalo herds that are endemic for bTB usually require tests with high sensitivity, since the aim is to identify as many M. bovis infected buffaloes as possible, at the expense of culling some false-positive animals. A herd that is historically bTB-free or has a low bTB prevalence requires a test with high specificity, which will correctly identify M. bovis uninfected buffaloes while minimising the removal of false-positive animals.”

Unfortunately, since the abstract section has a maximum 200 word count, it was not possible to expand on all the topics in this section. We gave a brief description of the most important aspects of the paper and expanded on this throughout the other sections of the paper.

Comment 2:

Abstract. Before using “M.” as an abbreviation of Mycobacterium, you should mention the word in full followed by its abbreviation within brackets.

Response:

The word ”Mycobacterium” was introduced in line 16 of the abstract. We have added “M. bovis” in brackets following “Mycobacterium bovis” in line 16 in the abstract, and line 36 of the introduction section.

Comment 3:

Abstract. “In this study, the specificities of three assays were determined using M. bovis unexposed herds, and a high specificity diagnostic algorithm developed.” Mention what was your reference gold standard or used method for measuring specificity of your assays?

Response:

Since the buffaloes in this study were not culled, we did not have access to tissue samples for mycobacterial culture from these animals, which is considered the gold standard of mycobacterial infection. However, we used buffaloes from historically bTB-free herds; M. bovis were not detected by mycobacterial culture from oronasal swabs collected from these animals.

We determined the specificity using the following method, as described in the methods section (lines 211-215): “The specificities for individual tests and test combinations of the Mabtech IGRA, IPRA, and SCITT (in series and in parallel) were calculated based on the percentage of M. bovis unexposed buffaloes from historically bTB-free herds (Cohort 1) that were test negative on each test, using an online tool (MedCalc Software Ltd 2022; Ostend, Belgium).”

Unfortunately, since the abstract section has a maximum 200 word count, it was not possible to expand on all the topics in this section. We gave a brief description of the most important aspects in the paper and expanded on this throughout the other sections of the paper.

Comment 4:

Abstract. “Serial test interpretation using the IGRA and IPRA showed significantly greater specificity (98.3%) than individual tests or parallel testing (73%). When the SCITT was added, the algorithm had a 100% specificity” illustrate what does you mean by serial test interpretation? Add the specificity of each of IGRA, IPRA and SCITT separately, so that the advantage of higher specificity of using these tests in that way is more clear than using each of these tests alone.

Response:

We describe serial and parallel testing in lines 172-175: “The parallel interpretation of the IGRA and IPRA was considered test positive if the result was positive for either IGRA or IPRA (negative if both IGRA and IPRA negative), whereas a positive serial test result was assigned if both IGRA and IPRA were positive, or positive if all three tests were positive (based on IGRA, IPRA and SCITT).” We have added “positive on both” in line 19 in the abstract, so that the line reads: “Serial test interpretation (positive on both) using the IGRA and IPRA showed significantly greater specificity….”.

We have added the individual specificities in line 20, so that it reads: “…using the IGRA and IPRA showed significantly greater specificity (98.3%) than individual (90.4% and 80.9%, respectively) tests or parallel testing (73%).”

Comment 5:

Mention the method used for determination of specificity in the abstract.

Response:

We determined the specificity using the following method, as described in the methods section (lines 211-215): “The specificities for individual tests and test combinations of the Mabtech IGRA, IPRA, and SCITT (in series and in parallel) were calculated based on the percentage of M. bovis unexposed buffaloes from historically bTB-free herds (Cohort 1) that were test negative on each test, using an online tool (MedCalc Software Ltd 2022; Ostend, Belgium).”

Unfortunately, since the abstract section has a maximum 200 word count, it was not possible to expand on all the topics in this section. We have stated in the abstract, lines 17-19: “In this study, the specificities of three assays were determined using M. bovis- unexposed herds, historically negative, and a high specificity diagnostic algorithm developed.” We gave a brief description of the most important aspects in the paper and expanded on this throughout the other sections of the paper.

Comment 6:

Abstract. “No association was found between NTM presence (in oronasal swab cultures) and positive cytokine assay results.” What about NTM which may exist in other parts of the body?

Response:

This study only investigated NTMs present in buffalo oronasal secretions. We therefore cannot comment on NTMs present in tissues or other secretions, such as urine, faecal or tissue samples.

Comment 7:

Abstract. Add more details about the used methodology within the abstract.

Response:

Unfortunately, since the abstract section has a maximum 200 word count, it was not possible to expand on all the topics in this section. We gave a brief description of the most important aspects of the paper and expanded on this throughout the other sections of the paper.

Comment 8:

Abstract. Mention what does you mean by “serial testing”?

Response:

Please see response in comment 4.

Comment 9:

Abstract. What does you mean by “high-value buffaloes”.

Response:

African buffaloes have high economic value in South Africa, and therefore we referred to them as “high-value buffaloes”. However, we have now removed this term from lines 24 – 25.

Comment 10:

Abstract. “Buffaloes positive on a single test (n=28) were regarded as test negative” what does you mean by single test? Do you mean either IGRA, IPRA or SCITT? Or do you mean that you had repeated testing by the same test several times?

Response:

Serial interpretation of bTB tests regards an animal as test positive when all tests included in the interpretation are positive. We applied IGRA and IPRA in series as a proof-of-principle to Cohort 2, and buffaloes were therefore test positive if positive on both. Buffaloes were regarded test negative if negative on either test (a single test) or on both tests. All tests (IGRA, IPRA, SCITT) were performed at the same time point.

Comment 11:

Abstract. “and M. bovis infection was confirmed following culling” what was the method of confirmation?

Response:

Buffaloes from Cohort 2 that were positive on both IGRA and IPRA were culled. We have changed wording in lines 26-27: “Four buffaloes were positive on IGRA and IPRA, and M. bovis infection was confirmed by culture.” Tissue samples were collected during post-mortem examination and mycobacterial culture and RD PCR sequencing was performed to confirm M. bovis infection in the buffaloes. Methods of mycobacterial culture and speciation were described in lines 178-187: “Oronasal swab samples (Cohort 1) and tissue samples from culled buffaloes (Cohort 2), were processed for mycobacterial culture in the biosafety level 3 (BSL-3) laboratory at Stellenbosch University. A 500 µl aliquot of each oronasal swab saline media, or approximately 1 cm3 of homogenised tissue (head, peripheral, abdominal and thoracic lymph nodes, lung) was cultured using the BACTECTM MGITTM 960 TB System (BD, Franklin Lakes, New Jersey, USA), as previously described. Samples were regarded as culture-negative if no growth was detected after 56 days of incubation; no downstream analysis was performed on these samples. Culture-positive samples were subjected to inactivation at 99°C for 30 min before removal from the BSL-3 facility for speciation”; and lines 204-206: “Mycobacterial cultures from tissue samples collected from culled Cohort 2 buffaloes during post-mortem examination were speciated by region of difference (RD) PCR, as described by Warren et al.”

Comment 12:

Abstract. “These results demonstrate the value of using IGRA and IPRA in series” illustrate what does you mean by “in series”?

Response:

“In series” referred to serial testing, as described above in comments 4, 8 and 10.

Comment 13:

Introduction. “The causative agent of bTB in buffaloes is Mycobacterium bovis, a pathogenic member of the Mycobacterium tuberculosis complex (MTBC)” mention that Mycobacterium bovis is the major cause of bovine tuberculosis, but Mycobacterium tuberculosis was reported recently to cause tuberculosis in buffalo and cite this paper “Borham, M., Oreiby, A., El‐Gedawy, A., Hegazy, Y., Hemedan, A., & Al‐Gaabary, M. (2022). Abattoir survey of bovine tuberculosis in tanta, centre of the Nile delta, with in silico analysis of gene mutations and protein–protein interactions of the involved mycobacteria. Transboundary and Emerging Diseases69(2), 434-450.”

Response:

We thank the reviewer for the suggestion. We have added the suggested reference; lines 35-38 in the introduction now read: “The primary causative agent of bTB in buffaloes is Mycobacterium bovis (M. bovis), a pathogenic member of the Mycobacterium tuberculosis complex (MTBC). However, other members of the MTBC have also been shown to cause TB in these bovids, including M. orygis and M. tuberculosis.”

Comment 14:

Introduction. “Since test-positive animals are usually culled, and the farm placed under quarantine until evidence can be provided that M. bovis has been eradicated from the herd, it is crucial to use accurate diagnostic tests to identify all infected animals without unnecessary culling of false-positive buffaloes.” Mention what is the evidence used in South Africa to consider that tuberculosis was eradicated from a farm after culling of infected cases and quarantine?

Response:

A herd is considered eradicated from disease if all animals in the herd have two consecutive negative TST test results, according to procedural policies by DALRRD (Bovine Tuberculosis manual 2016, and Veterinary Procedural Notice for buffalo disease risk management in South Africa 2017). We have added this in the introduction, lines 43-45 “A herd is considered eradicated from disease if all animals in the herd have two consecutive negative tuberculin skin test (TST) results.”  

Comment 15:

Introduction. “The inclusion of avian PPD to these tests has led to increased specificity by accounting for sensitisation to M. avium and closely related nontuberculous mycobacterial (NTM) species” Do you mean the inclusion of avian PPD in a comparative way with other PPD or antigens? Illustrate. Do not use the full term of NTM after being abbreviated before.

Response:

We have changed lines 75-82 in the introduction to read: “The single intradermal test measures the host response to bovine PPD. However, the inclusion of avian PPD in the SCITT has led to increased specificity by accounting for sensitisation to M. avium and closely related nontuberculous mycobacterial (NTM) species. In this test, the host immune response to bovine PPD is measured as a change in skin fold thickness at 72 hours. This is compared to the change in skin fold thickness in response to avian PPD. The difference in these results is then used to determine the test result of the buffalo.”

Nontuberculous mycobacteria were introduced for the first time in the introduction in line 78 and was therefore written out in full, followed by the abbreviation (NTM).

Comment 16:

Introduction. “Although current guidelines for testing buffaloes in South Africa rely on the SCITT, the use of parallel testing has been shown to improve detection of infected animals [4,9,14]” add more illustration about the parallel testing meant here?

Response:

Parallel and serial testing methods that were applied in this paper have been discussed in the Materials and Methods section, lines 172-175: “The parallel interpretation of the IGRA and IPRA was considered test positive if the result was positive for either IGRA or IPRA (negative if both IGRA and IPRA negative), whereas a positive serial test result was assigned if both IGRA and IPRA were positive, or positive if all three tests were positive (based on IGRA, IPRA and SCITT).”

Additionally, we included a sentence in the introduction section, lines 92-94 that now reads: “Although current guidelines for testing buffaloes in South Africa rely on the SCITT, the use of parallel testing (i.e., test-positive if positive on any bTB test included in the test algorithm) has been shown to improve detection of infected animals.”

Comment 17:

Introduction. “apply the high specificity testing algorithm to an unrelated historically bTB-free herd as a proof- of-principle.” What does you mean by the word unrelated?

Response:

Here we referred to Cohort 2, which was unrelated to the herds included in Cohort 1, i.e., Cohort 1 and Cohort 2 were not from the same buffalo herds. We have changed this word to independent in line 109; lines 104-110: “Therefore, this study aimed to 1) determine individual specificities of the QFT Mabtech IGRA, QFT IPRA, and SCITT in known M. bovis unexposed African buffaloes, 2) evaluate test algorithms to optimise specificity in these M. bovis unexposed buffalo herds, 3) determine if there is an association between false-positive bTB test results in M. bovis unexposed herds and presence of NTMs, and 4) apply the high specificity testing algorithm to an independent historically bTB-free herd as a proof-of-principle.”

Comment 18:

M&M. “Samples were opportunistically collected” illustrate what does you mean by opportunistically?

Response:

We removed the word “opportunistically” from line 113.

Comment 19:

M&M. Illustrate how the historically bTB-free private game farms were confirmed to be free? Does they usually subjected to regular periodical testing?

Response:

It is mandatory to test buffaloes for bTB before moving to other farms. Buffaloes that are test-positive, are usually culled. Buffaloes in these herds were regarded as historically bTB-free, since animals have been TST negative during regular periodic testing; TST-positive buffaloes were culled, and mycobacterial culture and speciation were performed on tissue samples collected during post-mortem examination. However, M. bovis has not been detected to date in these animals, although NTMs have been identified in some of these animals. This was mentioned in lines 331-336: “The specificity values of the Mabtech IGRA, IPRA, and SCITT, were calculated in a cohort of buffaloes that had no history of exposure to M. bovis, based on previous herd test results. Buffaloes from these herds had undergone sporadic culling and no M. bovis had been detected by mycobacterial culture of tissue samples. Although ante-mortem culture has limited sensitivity, no MTBC members were detected in the oronasal secretions of the unexposed buffaloes used to calculate specificity.”

We have additionally added a section in the materials and methods section, lines 113-118: “Samples were collected between 2017 and 2021 from African buffaloes (n = 268) at historically bTB-free private game farms in South Africa (Limpopo, North West and Northern Cape provinces). These farms were regarded as historically bTB-free based on negative TST results during periodic testing. Any TST-positive buffaloes were culled, and mycobacterial culture and speciation performed on tissue samples. However, M. bovis has not been detected to date in these animals.”

Comment 20:

M&M. Add more description about the geographical distribution of sampling areas.

Response:

We have added the South African provinces from where buffaloes were sampled in the materials and methods section, lines 113-115, which now read: “Samples were collected between 2017 and 2021 from African buffaloes (n = 268) at historically bTB-free private game farms in South Africa (Limpopo, North West and Northern Cape Provinces)”.

Comment 21:

M&M. “Cohort 1 consisted of 115 buffaloes from three historically bTB-free farms tested in 2021. The QFT Mabtech IGRA and QFT IPRA were performed on all buffaloes, and SCITTs were performed on a subset (n = 69).” Illustrate the chronological order of these tests, did you performed SCITTs before, after or simultaneously with IGRA and IPRA?

Response:

Whole blood was collected before the SCITT was performed. Following capture and immobilisation of buffaloes, SCITTs were performed, whole blood was collected for IGRA and IPRA, and swabs were collected.

We changed lines 152-153, which now read “Single comparative intradermal tuberculin tests (SCITT) were performed following immobilisation on a subset of Cohort 1 buffaloes (n = 69)”; and lines 118-121: “Following capture and immobilisation of buffaloes, whole blood was collected by jugular venipuncture into lithium heparin tubes, as previously described. The SCITT was performed after blood collection.”

Comment 22:

M&M: Did you considered oronasal culture negative swaps to be true negative? As you now there are many other forms of TB rather than the respiratory form, and even NTM may affect other parts of the body, and this is may affect your specificity calculations.

Response:

No, an animal with a culture negative oronasal swab may not necessarily indicate an uninfected buffalo, since an infected buffalo may not be shedding at the time of sampling. We acknowledge that NTMs may be present in other secretions from buffaloes, such as faecal, urine and tissue samples. However, for this study we only had respiratory secretion samples (oronasal swabs) available and could therefore only base our findings on NTMs present in oronasal swabs.

Comment 23:

M&M. Does the aim of dividing the animals to cohort 1 and 2, is to compare the performance of the tests in TB-free and TB-containing populations?  If so, add more details about TB-positive cases of cohort 2.

Response:

Yes, the aim was to compare the bTB test performance in two independent groups of animals. Cohort 1 buffaloes were used to calculate the specificity of the SCITT, IGRA and IPRA, and to determine the specificity of these tests applied in parallel and in series.

Cohort 2 buffaloes came from a different herd and location to Cohort 1 and was therefore unrelated. This cohort was used as a test group to apply the proof-of-principle high specificity test algorithm to determine if this test interpretation would be accurate for M. bovis detection in historically bTB-free herds.

In the materials and methods section, we described the Cohort that was used, lines 138-144: ”Cohort 2 consisted of 153 buffaloes from one historically bTB-free farm (different location to Cohort 1). A single SCITT reactor buffalo bull was detected during bTB testing for movement for sale, according to regulations by the South African Department of Agriculture, Land Reform and Rural Development (DALRRD). Following this, the entire herd was tested in 2017, using QFT Cattletype® IGRA and QFT IPRA. The owner consented to have the buffaloes that had positive results on both assays culled, and tissue samples were collected during post-mortem examination for mycobacterial culture.” We revealed all results for this group in the results and discussion sections.

Bovine TB positive cases in Cohort 2 were discussed in both the results, lines 284-295: “Based on the high specificity of the combined Mabtech IGRA and IPRA when used in series, this algorithm was selected for retrospective evaluation of results from an unrelated historically bTB-free buffalo herd (n = 153; Cohort 2). Although the Cattletype® IGRA was used for Cohort 2 buffaloes, Cattletype® and Mabtech IGRAs had similar test performances. Twenty-eight of 153 buffaloes (18%) had positive IPRA results but were negative on Cattletype® IGRA. Using the serial interpretation, these animals were considered test negative. Four buffaloes (3%) were positive for both Cattletype® IGRA and IPRA and considered test positive. These four individuals had been culled due to suspicion of being infected, and post-mortem examinations performed. All four buffaloes were confirmed to be infected with M. bovis based on mycobacterial culture of tissues and speciation by RD PCR. The remaining 124 buffaloes had negative results for both Cattletype® IGRA and IPRA”, and discussion, lines 386-403: “As a proof-of-principle, the high specificity test algorithm with CRAs showed good diagnostic performance when applied to a historically bTB-free buffalo herd (Cohort 2). Although the specificity calculation for QFT IGRA was performed using the Mabtech bovine IFN-γ ELISA Pro assay, the buffaloes in the test group used the Cattletype® Ruminant IGRA. However, no significant differences in test performance between these assays have been reported. The Cattletype® IGRA was recently discontinued by the manufacturer, and therefore Mabtech IGRA was performed for Cohort 1 buffaloes (Cohort 2 buffaloes were tested before Cohort 1 animals). Even though a strict interpretation was used to define bTB test-positive individuals, four positive buffaloes were identified, which were confirmed to have M. bovis infection. Although some buffaloes were positive on IPRA, but not Cattletype® IGRA, it is possible that these animals had early infection, since IP-10 is released at greater concentrations than IFN-γ. However, infection could not be confirmed since these animals were not culled, due to absence of a compensation program, and the herd has been lost to follow-up. In this scenario, it is recommended that these buffaloes be quarantined and retested, since it is expected that truly infected animals will eventually become IGRA-positive. This example demonstrates the importance of having a herd history for M. bovis infection to determine whether a high specificity or high sensitivity testing approach should be used.”

Comment 24:

M&M. Under “Data analysis” illustrate what does you mean by in series and in parallel?

Response:

We added a section in data analyses that explains parallel and serial testing of buffaloes, lines 209-211: “Buffaloes were considered positive if positive on either IGRA or IPRA, using the parallel test interpretation, whereas serial test interpretation regarded buffaloes positive if positive on IGRA, IPRA and SCITT (if included).”

Comment 25:

M&M. Under “Data analysis, how could you determined NTM presence or absence, although culture negative results do not eliminate the presence of NTM in other regions of the body?

Response:

To clarify this, we added “in oronasal swab cultures” to lines 221-223, which now reads: “Contingency tables were used to determine the association between NTM presence or absence in oronasal swab cultures and the Mabtech IGRA, IPRA and SCITT test results…”

Comment 26:

Results. Illustrate what is meant by parallel and serial testing, preferably within the material and methods section.

Response:

Serial and parallel test interpretation was discussed in previous comments and in the Materials and Methods section, lines 172-175: “The parallel interpretation of the IGRA and IPRA was considered test positive if the result was positive for either IGRA or IPRA (negative if both IGRA and IPRA negative), whereas a positive serial test result was assigned if both IGRA and IPRA were positive, or positive if all three tests were positive (based on IGRA, IPRA and SCITT).”, and lines 209-211: “Buffaloes were considered positive if positive on either IGRA or IPRA, using the parallel test interpretation, whereas serial test interpretation regarded buffaloes positive if positive on IGRA, IPRA and SCITT (if included).”

Comment 27:

Results. How could you identify false positive and true negative?

Response:

Since Cohort 1 buffaloes originated from historically bTB-free buffalo herds, we expected all animals to be test-negative. Therefore, in this context, we regarded buffaloes that were test-negative as “true negative” and those with positive bTB test results as “false positive”.

We describe this in the data analyses section, lines 211-217: “The specificities for individual tests and test combinations of the Mabtech IGRA, IPRA, and SCITT (in series and in parallel) were calculated based on the percentage of M. bovis unexposed buffaloes from historically bTB-free herds (Cohort 1) that were test negative on each test, using an online tool (MedCalc Software Ltd 2022; Ostend, Belgium). Since buffaloes originated from historically bTB-free herds, animals were regarded as true negative if they were test-negative (on IGRA, IPRA or SCITT), and false-positive if test-positive (on IGRA, IPRA or SCITT).”

Comment 28:

Discussion: Do you think that 4 positive cases of Cohort 2 are enough to finally judge your principle of the high specificity of IGRA and IPRA serial testing? I suggest mentioning that is a limitation for you study and further investigations are still required to confirm this principle.

Response:

The purpose of applying a high specificity test algorithm to historically bTB-free, or low bTB prevalence buffalo herds, is to minimise the number of false-positive animals, while accurately detecting truly infected buffaloes in the herd. Since Cohort 2 was considered a historically bTB-free herd, we did not expect many test-positive (if any) buffaloes. Therefore, 4 of 153 buffaloes that were positive on both IGRA and IPRA (and confirmed M. bovis infected), compared to 28 of 153 that were positive on either IGRA or IPRA, shows how specific this test interpretation is in accurately identifying truly infected buffaloes, without the loss of false-positive animals. A limitation of Cohort 2 was that we could not follow up on buffaloes that were positive on a single test, to determine if any of these buffaloes would have become test-positive on both tests during later stages of infection. This was discussed in lines 397-401: “However, infection could not be confirmed since these animals were not culled, due to absence of a compensation program, and the herd has been lost to follow-up. In this scenario, it is recommended that these buffaloes be quarantined and retested, since it is expected that truly infected animals will eventually become IGRA-positive.”

Comment 29:

Discussion: Add an interpretation why the presence of NTM showed no clear association with a positive CRA result?

Response:

The antigens used in the cytokine release assays, i.e., ESAT-6 and CFP-10, are highly specific to MTBC. Only a small number of NTMs have been identified to contain the genetic regions encoding these proteins, which may cause cross-reactive immune responses if present in the buffaloes. However, we could not confirm infection with NTMs, since we identified NTMs from oronasal secretions, rather than tissue samples, we could not draw definitive conclusions regarding the association with NTM infection and CRA results. Other studies have also suggested cross-reactivity with other bacterial species with antigens used in bTB diagnostic tests, which may have contributed to the false-positive CRA results that were observed in some of our buffaloes.

We added this information to the discussion section, lines 377-385: “The presence of Mycobacterium avium complex (MAC) members also did not appear to influence CRA results. Positive CRA results could also have been an indication of M. bovis infection, although MTBC was not detected in oronasal swab cultures. However, since oronasal swab cultures indicates the presence of NTMs in the oronasal cavities of the animals, it may not represent NTM infection. Other studies have also reported possible cross-reactivity with organisms other than NTMs, such as Nocardia, Rhodococcus and Corynebacterium, which are closely related to the Mycobacterium genus. Therefore, the cause of false-positive CRA results should be investigated in future studies.”

References:

Bernitz N, Goosen WJ, Clarke C, Kerr TJ, Higgitt R, Roos EO, Cooper DV, Warren RM, van Helden PD, Parsons SDC, et al. 2018. Parallel testing increases detection of Mycobacterium bovis-infected African buffaloes (Syncerus caffer). Vet Immunol Immunopathol 204.

Bernitz N, Kerr T, Goosen W, Chileshe J, Higgit R, Roos E, Meiring C, Gumbo R, de Waal C, Clarke C, et al. 2021. Review of Diagnostic Tests for Detection of Mycobacterium bovis Infection in South African Wildlife. Front Vet Sci. Front Vet Sci 8. https://pubmed.ncbi.nlm.nih.gov/33585615/. Accessed September 2021.

Bernitz N, Kerr TJ, Goosen WJ, Clarke C, Higgitt R, Roos EO, Cooper DV, Warren RM, van Helden PD, Parsons SDC, et al. 2019. Parallel measurement of IFN-γ and IP-10 in QuantiFERON®-TB Gold (QFT)plasma improves the detection of Mycobacterium bovis infection in African buffaloes (Syncerus caffer). Prev Vet Med 169.

Goosen WJ, Cooper D, Miller MA, Van Helden PD, Parsons SDC. 2015. IP-10 is a sensitive biomarker of antigen recognition in whole-blood stimulation assays used for the diagnosis of <i>Mycobacterium bovis<i> infection in African buffaloes (Syncerus caffer). Clin Vaccine Immunol 22:974–978.

Lyashchenko KP, Sridhara AA, Johnathan-Lee A, Sikar-Gang A, Lambotte P, Esfandiari J, Bernitz N, Kerr TJ, Miller MA, Waters WR. 2020. Differential antigen recognition by serum antibodies from three bovid hosts of Mycobacterium bovis infection. Comp Immunol Microbiol Infect Dis 69:101424.